# Learning Progress Driven Multi-Agent Curriculum

**Wenshuai Zhao** [1]   **Zhiyuan Li** [1]   **Joni Pajarinen** [1]

## Abstract

The number of agents can be an effective curriculum variable for controlling the difficulty of multi-agent reinforcement learning (MARL) tasks. Existing work typically uses manually defined curricula such as linear schemes. We identify two potential flaws while applying existing reward-based automatic curriculum learning methods in MARL: (1) The expected episode return used to measure task difficulty has high variance; (2) Credit assignment difficulty can be exacerbated in tasks where increasing the number of agents yields higher returns which is common in many MARL tasks. To address these issues, we propose to control the curriculum by using a TD-error based *learning progress* measure and by letting the curriculum proceed from an initial context distribution to the final task specific one. Since our approach maintains a distribution over the number of agents and measures learning progress rather than absolute performance, which often increases with the number of agents, we alleviate problem (2). Moreover, the learning progress measure naturally alleviates problem (1) by aggregating returns. In three challenging sparse-reward MARL benchmarks, our approach outperforms state-of-the-art baselines.

## 1. Introduction

Curriculum reinforcement learning (CRL) (Portelas et al., 2020b) consists of a teacher generating a sequence of tasks of varying difficulty to train agents in order to bypass the difficulty of exploration in target tasks with sparse rewards. In single-agent CRL, the teacher usually controls the initial states or environmental parameters to change the task difficulty. When considering curriculum learning in multi-agent reinforcement learning (MARL) settings (Li et al., 2025; Zhao et al., 2024; Li et al., 2024), it is natural to employ

the number of agents as a potential curriculum in addition to the environmental parameters in single-agent CRL, as the number of agents is critical for the difficulty of MARL tasks. However, prior work is limited to controlling the number of agents manually or heuristically, for example, FewToMore (Wang et al., 2020) and EPC (Long et al., 2020) simply train the agents starting from tasks with fewer agents and gradually move to the target task in a predefined way. Variational automatic curriculum learning (VACL) (Chen et al., 2021) proposes a general automatic CRL method that can control both the initial states and the number of agents. However, the number of agents is still manually set in a linear scheme.

Existing works that assume a curriculum proceeding from fewer to more agents (Wang et al., 2020; Long et al., 2020) are typically based on the intuition that coordinating a larger number of agents is inherently more challenging than coordinating fewer agents to complete a task (Lowe et al., 2017; Rashid et al., 2018; Chao et al., 2021). However, this assumption does not always hold. For example, in the MPE *Simple-Spread* (Lowe et al., 2017) task shown in Figure 1, several agents (blue circles) try to cover as many landmarks (red circles) as possible. Intuitively, if the number of agents increases sufficiently, the most naive policy such as random moving can work well and achieve high episode returns. On the contrary, with fewer agents, the agents have to learn a complicated cooperation strategy to accomplish the task. Moreover, while an increased number of agents can lead to higher returns, this may also exacerbate credit assignment challenges in policy learning. We argue in this paper that more sophisticated control of the number of agents is needed in multi-agent curriculum reinforcement learning.

Therefore, we first directly apply the state-of-the-art single-agent automatic curriculum reinforcement learning (ACRL) method, self-paced reinforcement learning (SPRL) (Klink et al., 2021), to adaptively control the number of agents. We name this method as SPRLM. SPRLM explores a range of tasks with different contexts (Schaul et al., 2015a) and seeks easier ones with higher performance while making progress towards the target task, hence generating a reasonable task sequence as a curriculum. In our experiments, SPRLM outperforms heuristic baselines by successfully generating effective task distributions without being restricted to predefined task sequences. However, current ACRL methods,

---

[1]Department of Electrical Engineering and Automation, Aalto University, Espoo, Finland. Correspondence to: Wenshuai Zhao <wenshuai.zhao@aalto.fi>.

*Proceedings of the 42$^{nd}$ International Conference on Machine Learning*, Vancouver, Canada. PMLR 267, 2025. Copyright 2025 by the author(s).

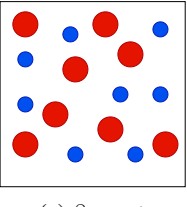
(a) 8 agents

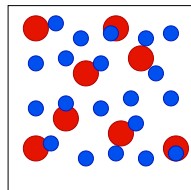
(b) 20 agents

*Figure 1.* In *Simple-Spread* task, agents (blue circles) need to cover as many landmarks (red circles) as possible. With the number of landmarks fixed, 20 agents shown on the **right** can easily complete the task and achieve higher returns compared to 8 agents on the **left**. However, a higher number of agents exacerbates the credit assignment problem in policy learning.

including SPRL, typically evaluate task difficulty based on expected episode returns, which introduces two issues in multi-agent settings. First, the episode returns of a task $c$, $G(c) = \sum_{t=0}^{T} \gamma^t r_t$, are usually sparse and can be only evaluated once per episode, leading to high variance in the estimated task difficulty (Sutton & Barto, 2018). Second, a curriculum focused on maximizing rewards may result in tasks that are ineffective for policy improvement. For instance, in the extreme case of the *Simple-Spread* task with an infinite number of agents, the initial random policy may suffice to be optimal, resulting in a zero policy gradient. In many Dec-POMDP (Amato et al., 2013) tasks where agents share the same reward, increasing the number of agents can lead to higher returns while also complicating credit assignment.

Inspired by the above findings, we further introduce self-paced multi-agent reinforcement learning (SPMARL). SPMARL tackles the two issues of high variance returns and complex credit assignment with many agents by optimizing a novel objective based on *learning progress* instead of expected episode returns. We use the expected *temporal difference* (TD) error to measure *learning progress*, that is, the *value loss* in multi-agent reinforcement learning. This new approach leverages the advantageous properties of the value function $V^\pi(s)$. As $V^\pi(s)$ reflects the performance of the current policy, the associated value loss w.r.t. different tasks naturally indicates the extent of policy changes on these tasks, given that converged value estimates typically imply no further policy updates. Moreover, the value loss is evaluated across all state transitions, significantly reducing estimation variance compared to episode returns.

We evaluate our methods on three benchmarks including MPE *Simple-Spread* (Lowe et al., 2017), the *XOR* matrix game (Fu et al., 2022), and four SMAC-v2 *Protoss* tasks (Ellis et al., 2022). Each benchmark is modified to introduce severe sparse rewards, posing significant exploration challenges. The results show that the number of agents can

serve as an effective curriculum variable to facilitate exploration. SPRLM outperforms heuristic baselines on most tasks, while SPMARL consistently surpasses all baselines, including other ACRL methods applied to MARL.

Our contributions are threefold: (1) SPRLM extends single-agent SPRL to a multi-agent context and shows that a principled curriculum based on the number of agents outperforms manually designed baselines; (2) We identify two flaws in the straightforward extension and propose SPMARL to address these issues; (3) Our experiments on three distinct benchmarks demonstrate that SPMARL outperforms baseline methods[1].

## 2. Related Work

In this section, we discuss general automatic curriculum generation methods and existing works that apply curriculum learning to multi-agent tasks, especially those employing the number of agents as a curriculum variable in order to scale up the current MARL methods.

**Automatic Curriculum Reinforcement Learning:** Curriculum learning has been extensively studied in the single-agent domain and sometimes in the context of unsupervised environment design (UED) (Teoh et al., 2024). A set of automatic curriculum generation methods are proposed (Portelas et al., 2020b) to control initial states (Florensa et al., 2017), goal positions (Florensa et al., 2018) or environment dynamics (Matiisen et al., 2019; Klink et al., 2021). Common objectives in the literature aim to optimize controllable contexts based on criteria such as *reward* (Colas et al., 2019; Klink et al., 2021) and *difficulty* (Florensa et al., 2018). Reward-based methods usually generate tasks with high returns, while the latter seeks tasks of intermediate difficulty rather than the easiest tasks. Several studies (Florensa et al., 2017; Mysore et al., 2019; Portelas et al., 2020a) employ the concept of *learning progress* to maximize training efficiency, similar to our approach. However, we note that these methods still measure the *learning progress* based on the sparse episode return increases, while our method estimates *learning progress* based on the *critic loss* from the underlying MARL updates which is much more stable. More importantly, the extensive literature on single-agent ACRL largely overlooks the credit assignment problem that emerges when using the number of agents as a curriculum variable in multi-agent settings. In the following, we simply refer to these ACRL methods as *reward-based* ACRL methods and build our work primarily on one representative ACRL method, SPRL (Klink et al., 2021). We argue that the improvements we propose in SPMARL can also be applied to other *reward-based* ACRL methods in the context of MARL tasks.

[1]Source Code: https://github.com/wenshuaizhao/spmarl

**Multi-Agent Curriculum Reinforcement Learning:** Compared to extensive study in the single-agent realm, only a few works have explored multi-agent curriculum reinforcement learning. Dynamic Multi-Agent Curriculum Learning (DyMA-CL) (Wang et al., 2020) solves large-scale problems by learning from a small-size multi-agent scenario and progressing to the target number of agents, where the number of agents is manually chosen. EPC (Long et al., 2020) expands the number of agents in the order of $N \rightarrow 2N$, training multiple parallel agents at each stage and selecting the best ones for the next stage through a crossover process. Variational Automatic Curriculum Learning (VACL) (Chen et al., 2021) presents a principal method for curriculum learning, framing the optimization of a curriculum distribution as a variational inference problem. Although VACL provides a general framework for controlling environment parameters, it adheres to a predetermined sequence $k' = k + 1$ or $k' = 2k$ for adjusting the number of agents. Consequently, we categorize these curriculum MARL approaches as a *Linear* baseline, incorporating both increasing (from fewer to more agents) and decreasing (from more to fewer agents) curricula in our experiments. Recently, (Wu et al., 2024) introduced a method for accounting for both performance and similarity to target tasks. However, the similarity is assessed using state visitation distributions, which are challenging to apply directly to tasks with differing state spaces as the number of agents varies.

## 3. Problem Formulation

In this section, we introduce the multi-agent reinforcement learning framework Dec-POMDP and the curriculum learning framework contextual reinforcement learning.

### 3.1. Dec-POMDP

We study the decentralized partially observable Markov decision process (Dec-POMDP) problem (Amato et al., 2013), which can be formulated as a tuple: $\langle \mathcal{S}, \{\mathcal{O}^i\}_{i \in \mathcal{N}}, \{\mathcal{A}^i\}_{i \in \mathcal{N}}, r, \mathcal{P}, \gamma \rangle$, where $\mathcal{N} = \{1, \cdots, n\}$ denotes a set of agents. At time step $t$ of, each agent $i$ observes local observation $o^i$ from the full state $s_t$ in the state space $\mathcal{S}$ of the environment and performs an action $a_t^i$ in the action space $\mathcal{A}^i$ based on its policy $\pi^i(\cdot|h^i)$, where $h^i$ encodes the history information of partial observations and actions. The joint policy consists of all the individual policies $\boldsymbol{\pi}(\cdot|s_t) = \pi^1 \times \cdots \times \pi^n$. The environment takes the joint action of all agents $\mathbf{a}_t = \{a_t^1, \cdots, a_t^n\}$, changes its state following the dynamics function $\mathcal{P} : \mathcal{S} \times \mathcal{A} \times \mathcal{S} \mapsto [0, 1]$ and generates a common reward $r : \mathcal{S} \times \mathcal{A} \mapsto \mathbb{R}$ for all the agents. $\gamma \in [0, 1)$ is a reward discount factor. The agents learn their individual policies and maximize the expected return: $\boldsymbol{\pi}^* = \arg\max_{\boldsymbol{\pi}} \mathbb{E}_{s, \mathbf{a} \sim \boldsymbol{\pi}, \mathcal{P}}[\sum_{t=0}^{\infty} \gamma^t r(s_t, \mathbf{a}_t)]$, where $\mathbf{a}_t$ is

the joint action at time step $t$ sampled from decentralized policies $\pi^i(\cdot|h^t)$.

### 3.2. Contextual Reinforcement Learning

Different from a typical Markov decision process (MDP) with fixed transition properties $\mathcal{M} = \langle \mathcal{S}, \mathcal{A}, \mathcal{P}, r, \mathcal{P}_0 \rangle$, contextual reinforcement learning (Neumann et al., 2011; Schaul et al., 2015a) parameterizes MDPs by a contextual parameter $\mathbf{c} \in \mathcal{C} \subseteq \mathbb{R}^m$ which can be certain environmental parameters, goals or initial states, while assuming a shared state-action space over these MDPs, $\mathcal{M}(\mathbf{c}) = \langle \mathcal{S}, \mathcal{A}, \mathcal{P}_{\mathbf{c}}, r_{\mathbf{c}}, \mathcal{P}_{0,\mathbf{c}} \rangle$. The objective of contextual RL is defined as: $\max_\theta J(\theta, \mu) = \max_\theta \mathbb{E}_{\mu(\mathbf{c}), \mathcal{P}_{0,\mathbf{c}}(\mathbf{s})}[V_\theta(\mathbf{s}, \mathbf{c})]$, where we have the target context distribution $\mathbf{c} \sim \mu(\mathbf{c})$ and initial state distribution $\mathbf{s} \sim \mathcal{P}_{0,\mathbf{c}}(\mathbf{s})$. The value function $V_\theta(\mathbf{s}, \mathbf{c})$ denotes the expected discounted return in states $\mathbf{s}$ and under the context $\mathbf{c}$ following the conditioned policy $\pi(\mathbf{a}|\mathbf{s}, \mathbf{c}, \theta)$. Generally, contextual RL aims to generalize behavior over different tasks by exploiting the continuation between MDPs. In curriculum learning, we usually set the target context distribution $\mu(\mathbf{c})$ as a Dirac delta function, since we are interested in solving one specific task with fixed context $\mathbf{c}$ while exploiting other tasks from different distributions $\nu(\mathbf{c})$ as curriculum.

## 4. SPRL for Multi-Agent Curriculum

SPRLM directly applies self-paced reinforcement learning (SPRL) (Klink et al., 2021) to control the number of agents as a curriculum to address hard exploration problems in sparse-reward tasks.

Similar to general homotopy optimization methods (Allgower & Georg, 2012), CRL works by assuming the continuation between tasks with varying contexts, i.e. the policy learned from one task can be a good initialization for another task. In tasks where directly learning from the target context $\mu(\mathbf{c})$ is difficult, i.e. the maximization of the following objective is hard due to sparse reward under the target contexts,

$$\max_\theta \mathbb{E}_{\mu(c)}[J(\theta, \mathbf{c})] = \max_\theta \mathbb{E}_{\mu(\mathbf{c}), \mathcal{P}_{0,\mathbf{c}}(\mathbf{s})}[V_\theta(\mathbf{s}, \mathbf{c})], \quad (1)$$

SPRL solves the hard exploration problem by first training the agent on easier tasks in contextual MDPs and then progressing to the target task. Formally, SPRL can be formulated as a constrained optimization problem

$$
\begin{aligned}
\min_\nu \quad & D_{\mathrm{KL}}(p(\mathbf{c}|\nu) \| \mu(\mathbf{c})) \\
\text{s.t.} \quad & \mathbb{E}_{p(\mathbf{c}|\nu)}[J(\theta, \mathbf{c})] \geq V_{\mathrm{LB}}, \\
& D_{\mathrm{KL}}(p(\mathbf{c}|\nu_{\mathbf{k}}) \| p(\mathbf{c}|\nu_{k+1})) \leq \epsilon,
\end{aligned}
\quad (2)
$$

where $\mu(\mathbf{c})$ is the target context distribution, usually set as a Dirac delta distribution. $p(\mathbf{c}|\nu)$ denotes the context distribu-

tion parameterized by $\nu$ such as the mean and variance of a Gaussian distribution. $\nu_{k+1}$ represents the context distribution to be updated. This distribution is supposed to generate easier tasks compared to the target task. We manually set a performance threshold $V_{\mathrm{LB}}$ in order to find task distributions with satisfying performance. In the first constraint in Equation 2, we maximize $\mathbb{E}_{p(\mathbf{c}|\nu)}[J(\theta, \mathbf{c})]$ by estimating the expected values over the new context distribution via importance sampling,

$$\max_{\nu_{k+1}} \frac{1}{M} \sum_{i=1}^{M} \frac{p(\mathbf{c}_i|\nu_{k+1})}{p(\mathbf{c}_i|\nu_k)} V_\theta(\mathbf{s}_{i,0}, \mathbf{c}_i), \qquad (3)$$

in which $M$ is the number of rollouts we collect. $V_\theta(\mathbf{s}_{i,0}, \mathbf{c}_i)$ represents the initial state value under current context $\mathbf{c}_i$ and indicates the difficulty of current task with $\mathbf{c}_i$. Usually, $V_\theta(\mathbf{s}_{i,0}, \mathbf{c}_i)$ is estimated by the sparse episode returns.

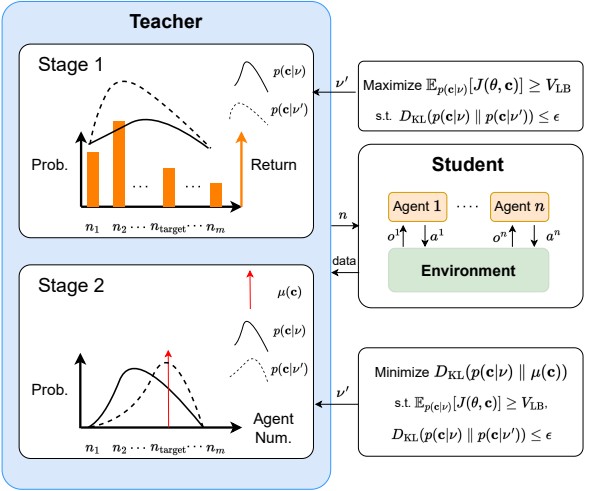

*Figure 2.* SPRL involves a two-stage optimization.

In practice, the constrained optimization problem in Equation 2 can be solved in a two-stage paradigm as shown in Figure 2. At first, when the expected performance $\mathbb{E}_{p(\mathbf{c}|\nu)}[J(\theta, \mathbf{c})]$ is lower than $V_{\mathrm{LB}}$, we maximize $\mathbb{E}_{p(\mathbf{c}|\nu)}[J(\theta, \mathbf{c})]$ under the constraint $D_{\mathrm{KL}}(p(\mathbf{c}|\nu_{\mathbf{k}}) \| p(\mathbf{c}|\nu_{k+1})) \leq \epsilon$ and obtain the new context distribution $p(\mathbf{c}|\nu_{k+1})$, which can be done by existing trust-region optimizer (Virtanen et al., 2020). Secondly, once the performance achieves the threshold $V_{\mathrm{LB}}$, we start to progress to the target context distribution $\mu(\mathbf{c})$ by minimizing $D_{\mathrm{KL}}(p(\mathbf{c}|\nu) \| \mu(\mathbf{c}))$ while still constrained by the KL divergence threshold $\epsilon$. Note that in this stage, SPRL maintains the performance always above $V_{\mathrm{LB}}$ by setting the context distribution unchanged until the performance is higher than the threshold again. In a multi-agent setting, SPRLM adaptively controls the number of agents as the context variable $\mathbf{c}$ to generate curriculum.

---

**Algorithm 1** Self-Paced Multi-Agent Reinforcement Learning (SPMARL)

---

1: **Input:** Initial context with number of agents distribution $\nu_0$, initial policy parameters $\theta_0$, target context with number of agents distribution $\mu(\mathbf{c})$, expected performance threshold $V_{\mathrm{LB}}$, number of iterations $K$, rollouts $M$ for each policy update, relative entropy bound $\epsilon$
2: **for** $k = 1$ **to** $K$ **do**
3:     **Policy Learning:**
4:     **for** $i = 1$ **to** $M$ **do**
5:         Sample context, i.e., the number of agents, $\mathbf{c}_i \sim p(\mathbf{c}|\nu_k)$
6:         Rollout trajectory $\tau_i \sim p(\mathbf{c}_i, \theta_k)$
7:     **end for**
8:     Obtain $\theta_{k+1}$ by the chosen MARL method and collected trajectories $\mathcal{D}_k = \{(\mathbf{c}_i, \tau_i) \mid i \in [1, M]\}$
9:     Estimate $V_{\theta_{k+1}}(\mathbf{s}_0^i, \mathbf{c}^i)$ for context $\mathbf{c}_i, i \in [1, M]$
10:    **Context Distribution Update:**
11:    *Stage 1: Optimize the **learning progress***
12:    **if** $\frac{1}{M} \sum_{i=1}^{M} V_{\theta_{k+1}}(\mathbf{s}_0^i, \mathbf{c}^i) < V_{\mathrm{LB}}$ **then**
13:        Obtain $\nu_{k+1}$ from Equation 5 under the KL divergence constraint
14:    *Stage 2: Progress to the target*
15:    **else**
16:        Obtain $\nu_{k+1}$ from Equation 2
17:    **end if**
18: **end for**

---

## 5. Self-Paced MARL

While SPRLM is supposed to be able to successfully find easier tasks with high performance and eventually mitigate the hard exploration problem due to sparse reward, the reward-based objective in SPRLM could lead to slow learning progress and unstable estimation. We improve SPRLM by proposing self-paced MARL (SPMARL) to optimize a new objective measuring the *learning progress* (LP) instead of directly optimizing task performance.

The primary requirement for the proposed concept of *learning progress* is that it should be an easily estimable measure of policy improvement. To this end, we draw inspiration from the advantageous properties of value function $V^\pi(s)$, which estimates the expected return of current policy $\pi_\theta$. Specifically, value loss inherently measures the extent of policy change over a particular task and can be approximated by the expected temporal difference (TD) error across all state transitions. Therefore, we posit that *value loss* can serve as an effective instantiation of *learning progress*, addressing the two key issues identified in reward-based CRL methods. In the SPMARL framework, the *value loss* is computed following the underlying MARL algorithm MAPPO (see appendix) defined as:

$$\mathrm{LP}(c) = \frac{1}{2}\mathbb{E}_{s,\mathbf{a}\sim\pi(\mathbf{a}|s,\mathbf{c})}[\|R(s,\mathbf{a}) - V(s)\|^2], \quad (4)$$

where $R(s,\mathbf{a})$ is the discounted return since state action pair $(s,\mathbf{a})$. Thanks to the CTDE framework allowing us to access the full state information during training, the estimation is sufficiently accurate. In SPMARL, we follow the two-stage optimization scheme in SPRL, but replace the reward objective of Equation 3 in the first stage with our new *learning progress* measurement

$$\max_{\nu_{k+1}} \frac{1}{M}\sum_{i=1}^{M} \frac{p(\mathbf{c}_i|\nu_{k+1})}{p(\mathbf{c}_i|\nu_k)}\mathrm{LP}_\theta(\mathbf{c}_i). \quad (5)$$

We keep the second stage optimization the same as SPRL, i.e. keeping the same performance threshold $V_{\mathrm{LB}}$. This is reasonable because even though we do not directly optimize returns over different contexts, the optimized *learning progress* objective will implicitly result in higher performance. As validated in our experiments, the curriculum generated by SPMARL even triggers a faster performance increase than SPRL. The detailed SPMARL is shown in Algorithm 1.

**Intuition on Learning Progress:** The idea of using TD error as the objective to generate tasks in SPMARL is analogous to the concept of Prioritized Experience Replay (PER) (Schaul et al., 2015b) in Deep Q-Networks (DQN) (Mnih et al., 2015), where samples with higher TD error are prioritized. However, unlike PER, SPMARL estimates the averaged TD error across different contexts and selects tasks with higher TD error, which are expected to enhance the policy learning progress.

## 6. Experiments

We evaluate our method on three challenging benchmarks with severe sparse rewards, including (1) MPE *Simple-Spread* task (Lowe et al., 2017) with 8 agents, (2) *XOR* game (Fu et al., 2022) with 20 agents, and (3) four SMAC-v2 *Protoss* tasks (Ellis et al., 2022). The specified number of agents pertains to the target task, while the curriculum learning methods generate a set of tasks with varying numbers of agents to train the policy for optimal performance on the target tasks. The hyper-parameters used in our experiments are listed in the appendix. We compare SPMARL with both SPRLM and several baselines:

- **SPRLM**: Our first algorithm, which directly applies SPRL (Klink et al., 2020) to MARL settings to adaptively control the number of agents. SPRLM also represents a set of general automatic CRL methods based on rewards (Portelas et al., 2020b) that can be applied to multi-agent settings.

- **Linear**: The linear scheme can be seen as an abstract of existing multi-agent curriculum learning works (Wang et al., 2020; Long et al., 2020; Chen et al., 2021). In *Simple-Spread* and *XOR* matrix game, we set a linearly increasing curriculum while in SMAC-v2 *Protoss 5 vs. 5* task we use a linearly decreasing scheme in order to diversify the baselines.

- **ALPGMM**: Absolute learning progress with Gaussian mixture models (ALPGMM) (Portelas et al., 2020a) uses Guassian mixture models to sample tasks with maximum absolute learning progress. However, the estimation of learning progress in ALPGMM is still based on differences in rewards, which may be prone to the same high variance issues that affect other reward-based curriculum learning methods.

- **VACL**: Variational Automatic Curriculum Learning (VACL) utilizes Stein Variational Gradient Descent (SVGD) to update the task distribution, prioritizing tasks associated with higher returns, akin to SPRL. However, unlike SPRL, VACL does not enforce strict convergence to the target task distribution, which can potentially hinder performance on the target tasks.

- **W/O teacher**: This is the default MARL algorithm MAPPO trained directly on the target task without any curriculum. In our experiments, due to the challenging sparse reward settings across all benchmarks, **W/O teacher** consistently fails on all tasks, yielding zero rewards. **Consequently, we exclude its performance from the figures for clarity.**

Note that ALPGMM was originally designed for single-agent tasks to optimize environmental parameters. In this work, we adapt it to dynamically control the number of agents. Similarly, while VACL employs a linear scheme for adjusting the number of agents in its original formulation, we extend its variational inference framework to enable adaptive agent control.

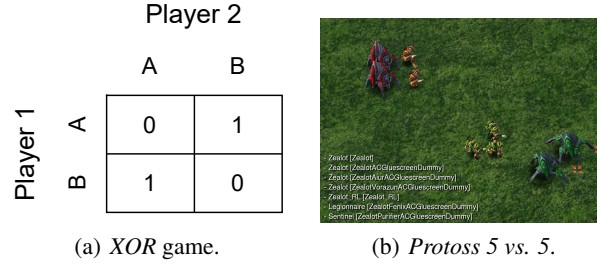

(a) *XOR* game.    (b) *Protoss 5 vs. 5*.

*Figure 3.* (a): The payoff matrix of 2-player *XOR* game. (b): Scenario from *Protoss 5 vs. 5* in SMACv2 showing agents battling the built-in AI.

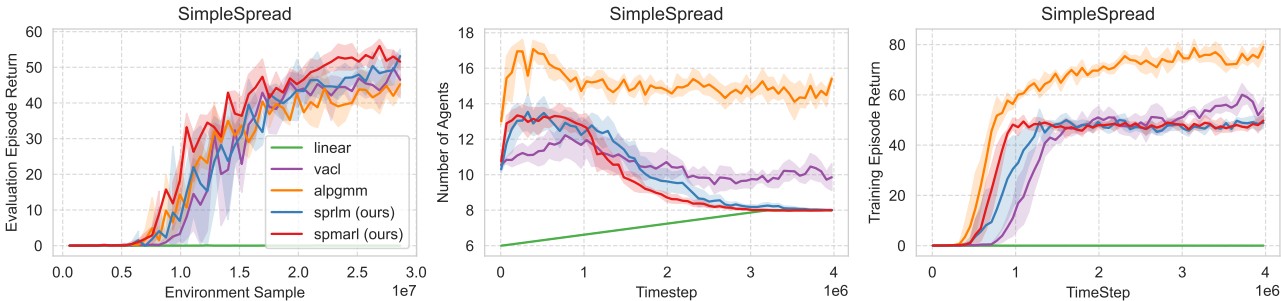

*Figure 4.* Comparison on the *Simple-Spread* task, where the target is set with 8 agents and 8 landmarks. The plots are averaged over 5 random seeds and the shadow area denotes the 95% confidence intervals. The **left** figure shows the evaluation returns on the target task with 8 agents. Note that the x-axis represents the samples collected from the environment, which is proportional to the number of agents. The **middle** figure presents the generated curriculum from different methods, where SPMARL and SPRLM first generate more agents and then converge to the target 8 agents while ALPGMM and VACL always generates more agents. The **right** figure shows the episode returns on the training tasks. The ALPGMM algorithm achieves the highest because it samples tasks with more than 14 agents.

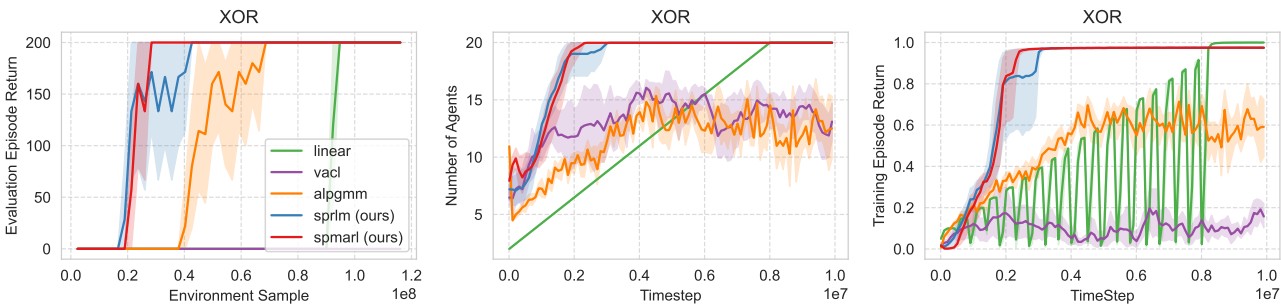

*Figure 5.* Comparison on the 20-player *XOR* game where each agent needs to output different actions to succeed. While the linear curriculum from few to more (*linear*) and *alpgmm* successfully achieve optima eventually, SPRLM and SPMARL demonstrate a faster convergence.

## 6.1. MPE Simple-Spread

As shown in Figure 1, *Simple-Spread* involves several agents trying to cover all the landmarks as soon as possible. The original *Simple-Spread* task is designed with a dense reward function denoted by the negation of summed distances to each landmark. However, we test our method on a modified version featuring sparse rewards which becomes challenging. In this modified version, agents can only receive rewards when at least 4 landmarks are covered and the reward is the number of landmarks successfully covered at each timestep. We set the target task with 8 agents and 8 landmarks.

**Analysis of the evaluation performance:** The experimental results are presented in Figure 4, demonstrating that the number of agents serves as an effective curriculum variable for addressing the sparse reward problem. In the left figure, which depicts episode returns evaluated on the target task during training, both SPMARL and SPRLM outperform

other baselines. SPMARL further surpasses SPRLM by achieving faster convergence, attributed to its more rapidly updated context distribution. ALPGMM generates a larger number of agents during training, leading to the highest training rewards. However, unlike the SPRL framework, it does not enforce convergence of the context distribution to the target distribution, resulting in lower evaluation performance. It is reasonable that *Linear* completely fails to learn any valid policies because of the severe sparse rewards. *Linear* generating curriculum from few to more agents even exacerbates the problem as in this task few agents have less chance to receive any non-zero rewards.

**Analysis of the generated curriculum:** It is noteworthy to observe the different curricula generated by SPMARL and SPRLM in the middle figure. SPMARL rapidly updates the context distribution to include more agents, whereas SPRLM continues to explore for a longer duration. As a result, SPMARL starts early to converge to the target task after achieving the performance threshold $V_{\text{LB}}$ of 50.

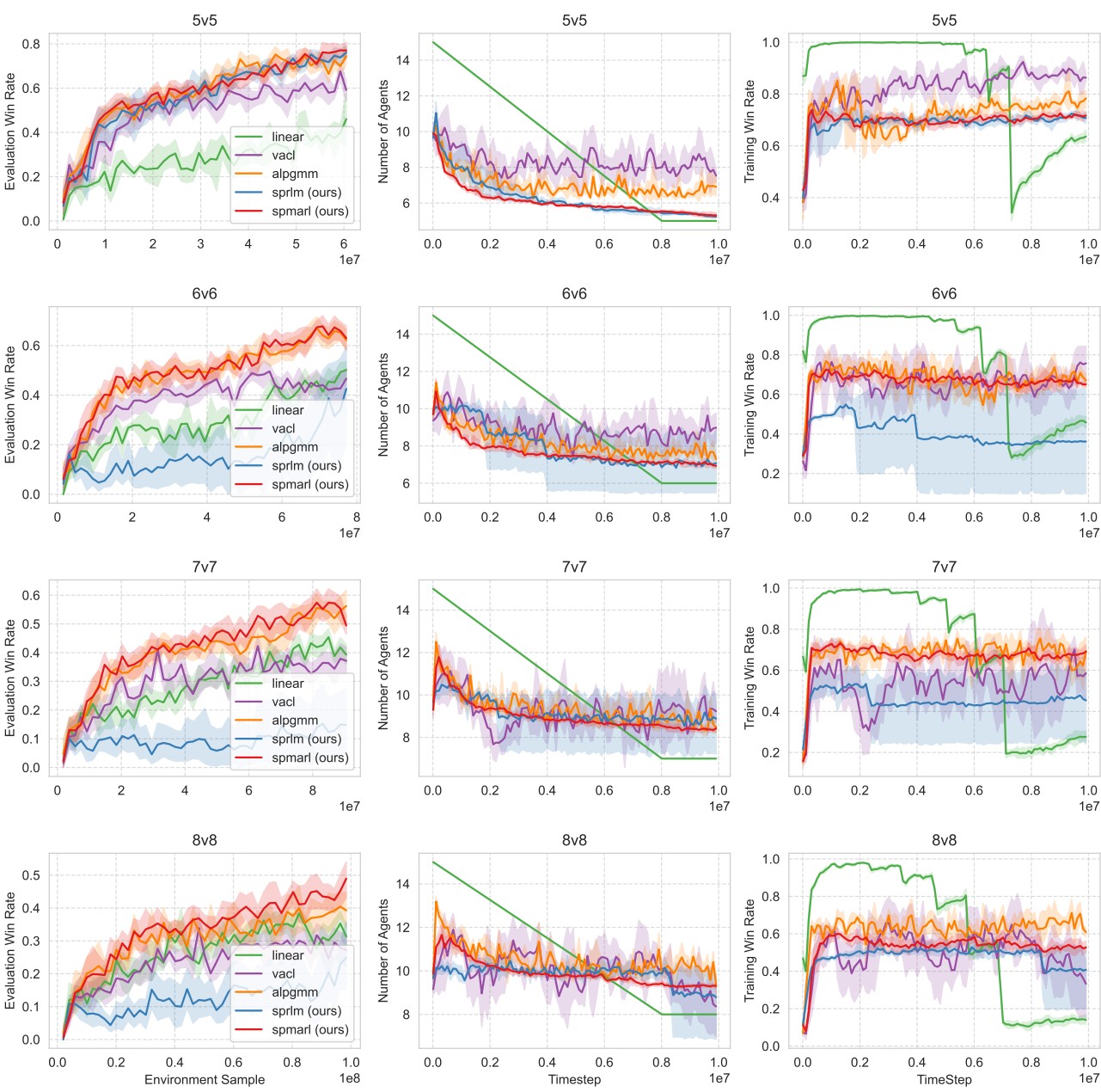

Figure 6. Comparison on SMACv2 *Protoss* tasks. From top to bottom row, the tasks are *5 vs. 5*, *6 vs. 6*, *7 vs. 7* and *8 vs. 8*. Across all four tasks, SPMARL achieves performance that is comparable to or better than all baseline methods.

The faster context updates in SPMARL highlight two key advantages provided by our proposed *learning progress* objective compared to the *performance* objective in SPRLM: more stable estimation and more effective task generation for policy improvement.

**Analysis of the training process:** The episode returns during training are closely influenced by the curriculum generated by different methods. As shown in the right figure,

ALPGMM achieves the highest training episode returns, as it consistently samples tasks with a larger number of agents without considering the target task. However, despite the apparent advantage of training with more agents, ALPGMM does not lead to improved performance on the target task. Similarly, VACL fails to converge to the target context distribution.

## 6.2. XOR Matrix Game

In this subsection, we conduct experiments on a cooperative task called $N$-player *XOR* game. In this game, each player has $N$ possible actions and they receive a positive reward only if they each select different actions. A simple example of 2-player *XOR* game is given by the payoff matrix shown in Figure 3(a). While learning the optimal policy for a 2-player *XOR* game is relatively simple, the complexity increases dramatically with the number of players due to the exponentially expanding joint policy space. Consequently, a curriculum learning approach becomes essential. In our experiments, we set the target task for 20 players.

As shown in Figure 5, VACL fails to learn an effective policy. The *Linear* baseline follows the human prior assumption that tasks with fewer agents are easier, gradually increasing the number of agents over time. While *Linear* eventually converges to the optimal policy, its need to explore the entire context space results in inefficiency and instability, as shown in the fluctuating training curves in the right figure. ALPGMM also converges to the optimal policy but only after extensive exploration, as its generated curriculum fails to converge to the target task.

In contrast, our methods automatically identify effective curricula. As shown in the middle figure, both SPRLM and SPMARL efficiently explore the context space and ultimately converge to the target 20 agents task. Notably, SPMARL further outperforms SPRLM by achieving faster convergence.

## 6.3. SMAC-v2

We further test our method on SMAC-v2 (Ellis et al., 2022), a new version of the StarCraft Multi-Agent Challenge (SMAC) (Samvelyan et al., 2019), which increases the difficulty by imposing higher stochasticity. Specifically, we evaluate SPMARL on the *Protoss* tasks (Figure 3(b)) with sparse reward. In this setting, the agents can only receive 1 if win or $-1$ if lose at the end of the game.

As illustrated in Figure 6, baseline methods incorporating curriculum learning successfully learn effective policies, highlighting the significance of curriculum learning for this task. Among these methods, SPMARL and ALPGMM outperform other curriculum learning approaches, namely *Linear* and *VACL*, in terms of both convergence speed and final performance. In contrast, SPRLM fails to generate an effective curriculum except for the *5v5* task, likely due to the extreme reward setting $\{-1, 0, 1\}$, which hinders the optimization of the curriculum distribution from converging to a viable solution.

It is notable that in the *Linear* baseline, even though with superior prior knowledge designing curriculum from sufficiently as high as 15 agents to the target number of agents, it

fails to show better evaluation performance on the target task despite much higher training performance in the right figure. These can be attributed to the exacerbated credit assignment difficulty due to too many agents. On the contrary, our method SPMARL learns to decrease the number of agents when the current context distribution has a sufficient number of agents to meet the performance threshold.

## 6.4. Comparison of the Objective Variances

We visualize the standard deviation (std.) of the objectives for SPMARL and SPRLM, specifically the std. of the TD error and episode return. As shown in Figure 7, SPMARL exhibits significantly lower variance compared to SPRLM in both *Protoss 7v7* and *8v8*. This observation further supports our hypothesis that a lower estimation variance in the TD error facilitates more effective curriculum generation. More results and significance analysis can be found in Section A.3.

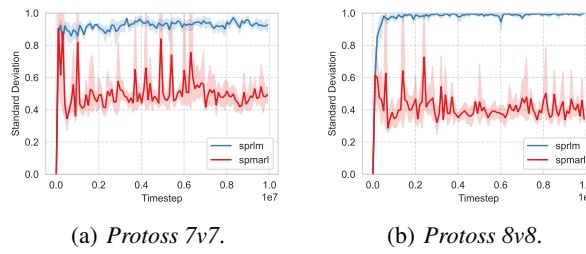

(a) *Protoss 7v7*.  (b) *Protoss 8v8*.

*Figure 7.* Comparison of the std. of the TD error objective used in SPMARL and the std. of the episode return used in SPRLM.

## 7. Discussion

We explore the potential of employing a controllable number of agents to alleviate the challenges inherent to multi-agent systems. Our second proposed method, SPMARL, enhances the convergence speed of the first straightforward algorithm SPRLM by introducing a novel *learning progress* objective but retains the two-stage optimization paradigm under the KL constraint, as shown in Figure 2. Although the KL constraint was initially proposed to maintain performance continuity during context updates in reward-based SPRL, SPMARL demonstrates superior performance improvements when not directly optimizing performance. While the KL constraint aids in stabilizing training and benefits the initial phase of this work, its necessity can be reconsidered when utilizing our *learning progress* as the new optimization objective. We leave this investigation as our future work.

## 8. Conclusion

In this paper, we investigate the control of agent numbers as an effective curriculum strategy. Existing works are typically limited to predefined curriculum that rely on heuris-

tics, such as a linear scheme. To address this limitation, we first directly extend the state-of-the-art single-agent curriculum method SPRL to the multi-agent setting. Our analysis reveals two potential flaws in general reward-based curriculum methods for MARL: unstable estimation based on sparse episode returns (Sutton & Barto, 2018), and the increased credit assignment difficulty in tasks where more agents tend to yield higher returns. These problems can eventually lead to slow learning progress. Therefore, we further propose SPMARL that prioritizes tasks based on *learning progress* instead of the episode returns. SPMARL optimizes *value loss* over the context distribution. Importantly, *value loss* naturally indicates policy improvement as tasks with higher *value loss* represents more significant policy change. Moreover, the expected *value loss* can be accurately estimated w.r.t. all the state transitions rather than sparse episode returns. Consequently, SPMARL generates tasks that benefit policy learning as much as possible. Although SPMARL does not focus directly on performance, it implicitly increases episode returns more effectively by improving *learning progress*. Evaluation on three challenging benchmarks demonstrates the effectiveness of SPMARL in addressing difficult sparse-reward problems. In experiments, SPMARL outperforms comparison methods.

## Acknowledgements

We acknowledge CSC – IT Center for Science, Finland, for awarding this project access to the LUMI supercomputer, owned by the EuroHPC Joint Undertaking, hosted by CSC (Finland) and the LUMI consortium through CSC. We also acknowledge the computational resources provided by the Aalto Science-IT project and funding by Research Council of Finland (357301). Zhiyuan Li is supported by the Research Council of Finland from the Flagship program: Finnish Center for Artificial Intelligence (FCAI).

## Impact Statement

This paper presents work whose goal is to advance the field of Machine Learning. There are many potential societal consequences of our work, none which we feel must be specifically highlighted here.

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

# A. Appendix

## A.1. Implementation Factors

Several implementation details need to be considered when applying curriculum learning to control the number of agents. First, the changing number of agents usually leads to varying vector lengths of agents' observation and we need to handle the new coming agents. Second, the context distribution in SPRL is usually implemented as a Gaussian distribution which is continuous while we need to optimize the discrete number of agents. Third, the existing MARL benchmarks rarely support setting different numbers of agents in a convenient way.

In SPMARL, we tackle the first difficulty by simply setting a fixed-length observation vector for each agent when it doesn't impact the performance much (Zhao et al., 2023) or padding zeros to the state vectors. To handle the newly joined agents, we adopt parameter sharing for all the agents so that the new agents can use the learned policy. For the second problem, we retain the Gaussian distribution and simply discretize the sampled float contexts, which works well in practice. In the third problem, we create a set of environment wrappers for existing MARL benchmarks to easily change the number of agents.

In addition, $V_{\mathrm{LB}}$ is an important hyper-parameter in SPRL and SPMARL. In our experiments, $V_{\mathrm{LB}}$ can be usually set as $80\%$ of the expected return on the target task. We perform an ablation in Section A.3.5.

## A.2. MAPPO

Multi-agent PPO (MAPPO) (Chao et al., 2021) is a popular MARL algorithm with strong performances on various benchmarks. MAPPO works in a straightforward way by applying single-agent PPO (Schulman et al., 2017) to multi-agent learning while using a centralized critic with additional full-state information. In MAPPO, each agent learns a centralized state value function $V(s)$, and the individual policy is updated by maximizing the following objective

$$\max_{\pi_{\theta^i}} \mathbb{E}_{(s,a^i)\sim\pi^i}[\min(r(\theta)A(s,a^i), \mathrm{clip}(r(\theta), 1\pm\epsilon)A(s,a^i))], \tag{6}$$

where $r(\theta)$ is the importance ratio between the current policy and the previous policy used to generate the data,

$$r(\theta) = \frac{\pi_{\theta^i}(a_t^i|h_t^i)}{\pi_{\theta_{\mathrm{old}}^i}(a_t^i|h_t^i)} \tag{7}$$

The advantage $A(s,a^i)$ is usually estimated by the generalized advantage estimator (GAE) (Schulman et al., 2015) defined as Equation 8, where we use the full state information $s$ thanks to the centralized training and decentralized execution (CTDE) (Lowe et al., 2017) framework,

$$A_t^{\mathrm{GAE}(\lambda,\gamma)} = \sum_{l=0}^{\infty} (\gamma\lambda)^l \delta_{t+l}, \tag{8}$$

and $\delta_t$ denotes the TD error

$$\delta_t = r_t + \gamma V(s_{t+1}) - V(s_t). \tag{9}$$

In our implementation, in order to transfer to different numbers of agents, we use parameter sharing (Chao et al., 2021) among agents.

## A.3. More Results

In this section, we present additional experimental results to provide a comprehensive evaluation and analysis of our method. Section A.3.1 reports a significance analysis of the experiments presented in the main text. We extend the empirical comparison to four additional SMAC-v2 tasks in Section A.3.2, and to two *BenchMARL* benchmarks (Bettini et al., 2024) in Section A.3.3. In Section A.3.4, we further compare the variance of SPMARL and SPRLM across four tasks. Finally, Section A.3.5 provides an ablation study on the hyperparameter $V_{\mathrm{LB}}$ using two SMAC-v2 tasks.

### A.3.1. SIGNIFICANCE ANALYSIS

In addition to the learning curves presented in the main text, we report the statistical results of the experiments in the following tables. Table 1 summarizes the mean evaluation performance of various algorithms during training, showing that SPMARL consistently outperforms the baselines or achieves comparable best-case results.

*Table 1.* Mean and standard deviation of the evaluation performance during training on all tasks. Results for methods compared to SPMARL with p-value higher than 0.05 according to the Mann-Whitney U test (see Table 2) are highlighted in bold.

| Algorithm | SimpleSpread | XOR | 5v5 | 6v6 | 7v7 | 8v8 |
|---|---|---|---|---|---|---|
| W/O Teacher | 0.00 (0.00) | 0.00 (0.00) | 0.00 (0.00) | 0.00 (0.00) | 0.00 (0.00) | 0.00 (0.00) |
| Linear | 0.00 (0.00) | 26.80 (1.16) | 0.25(0.02) | 0.25 (0.02) | 0.25 (0.01) | 0.25 (0.01) |
| VACL | 24.43 (4.53) | 0.00 (0.00) | 0.47 (0.02) | 0.37 (0.02) | 0.28 (0.01) | 0.22 (0.01) |
| ALPGMM | 24.65 (1.98) | 104.80 (14.73) | **0.55 (0.02)** | **0.45 (0.01)** | 0.36 (0.01) | 0.28 (0.01) |
| SPRLM | 27.48 (5.38) | **141.45 (16.98)** | **0.54 (0.05)** | 0.13 (0.15) | 0.08 (0.11) | 0.11 (0.07) |
| SPMARL | **33.36 (1.84)** | **143.08 (7.72)** | **0.55 (0.01)** | **0.45 (0.01)** | **0.38 (0.02)** | **0.31 (0.01)** |

To further assess the significance of these differences, we perform a one-sided Mann–Whitney U test (Mann & Whitney, 1947) to evaluate whether the evaluation returns of SPMARL are stochastically greater than those of the baselines. As shown in Table 2, the test yields p-values below 0.05 in most cases.

*Table 2.* Statistical significance (p-values) of SPMARL compared to baseline methods.

| Algorithm | SimpleSpread | XOR | 5v5 | 6v6 | 7v7 | 8v8 |
|---|---|---|---|---|---|---|
| Linear | 0.0060 | 0.0085 | 0.0079 | 0.0079 | 0.0040 | 0.0040 |
| VACL | 0.0040 | 0.0052 | 0.0040 | 0.0079 | 0.0079 | 0.0079 |
| ALPGMM | 0.0040 | 0.0097 | 0.7222 | 0.4524 | 0.0079 | 0.0317 |
| SPRLM | 0.0159 | 0.8395 | 0.4206 | 0.0079 | 0.0022 | 0.0040 |

### A.3.2. EXPERIMENTS ON MORE SMAC-V2 TASKS

We conduct additional experiments on four SMAC-v2 tasks: *Terran 5 vs. 5*, *Terran 6 vs. 6*, *Zerg 5 vs. 5*, and *Zerg 6 vs. 6*. As shown in Figure 8, SPMARL consistently outperforms baseline methods and generates coherent and effective curricula across all tasks.

### A.3.3. EXPERIMENTS ON *BenchMARL* TASKS

We further evaluate our method on two additional tasks from the recent *BenchMARL* benchmark (Bettini et al., 2024), beyond the already-included *SMAC-v2* and *Simple-Spread* environments. As shown in Figure 9, SPMARL achieves performance comparable to other baselines. It is important to note that the two new tasks, *Balance* and *Wheel*, employ the same dense-reward setting as in the original *BenchMARL*, which may reduce the presence of exploration challenges and thus diminish the benefits of curriculum learning. Nonetheless, SPMARL remains the only method that generates curricula which converge to the desired target distributions.

### A.3.4. VARIANCE COMPARISON ON MORE TASKS

**How to compute the variance:** Mathematically, assume we have collected a set of contexts $\{c_1, c_2, \ldots, c_n\}, n \approx 25$, the corresponding episode returns $\{R_1, R_2, \ldots, R_n\}$, the TD-errors $\{\text{TD}_1, \text{TD}_2, \ldots, \text{TD}_n\}$ are computed as $\text{TD}_i = \text{LP}(c_i) = \mathbb{E}_{s, \mathbf{a} \sim \pi(\mathbf{a}|s, c_i)} \left[ \| R(s, \mathbf{a}) - V(s) \|^2 \right]$. Note that the return used in computing TD-errors is usually estimated as the critic in RL, $R(s_t, \mathbf{a}_t) = r_t + \gamma V_{s_{t+1}}$, further reducing the estimation variance (Schulman et al., 2015). The standard deviation is then computed as $\sigma = \sqrt{\frac{1}{n} \sum_{i=1}^{n} (x_i - \bar{x})^2}$, where $x_i$ can be $R_i$ or $\text{TD}_i$ and $\bar{x}$ represents the mean. Since TD-error on each context has been averaged over the samples of the whole episode, it usually shows lower estimation variance.

We present additional results in Figure 10 comparing the standard deviation of objective estimations between SPMARL and SPRLM on four more tasks, further highlighting the stability of our proposed approach.

### A.3.5. ABLATION ON $V_{\text{LB}}$

The hyperparameter $V_{\text{LB}}$ plays a crucial role in SPRL methods (Klink et al., 2020; 2021). As our approach adopts the two-stage optimization framework from SPRL, we follow a similar strategy for setting $V_{\text{LB}}$. Specifically, in our experiments,

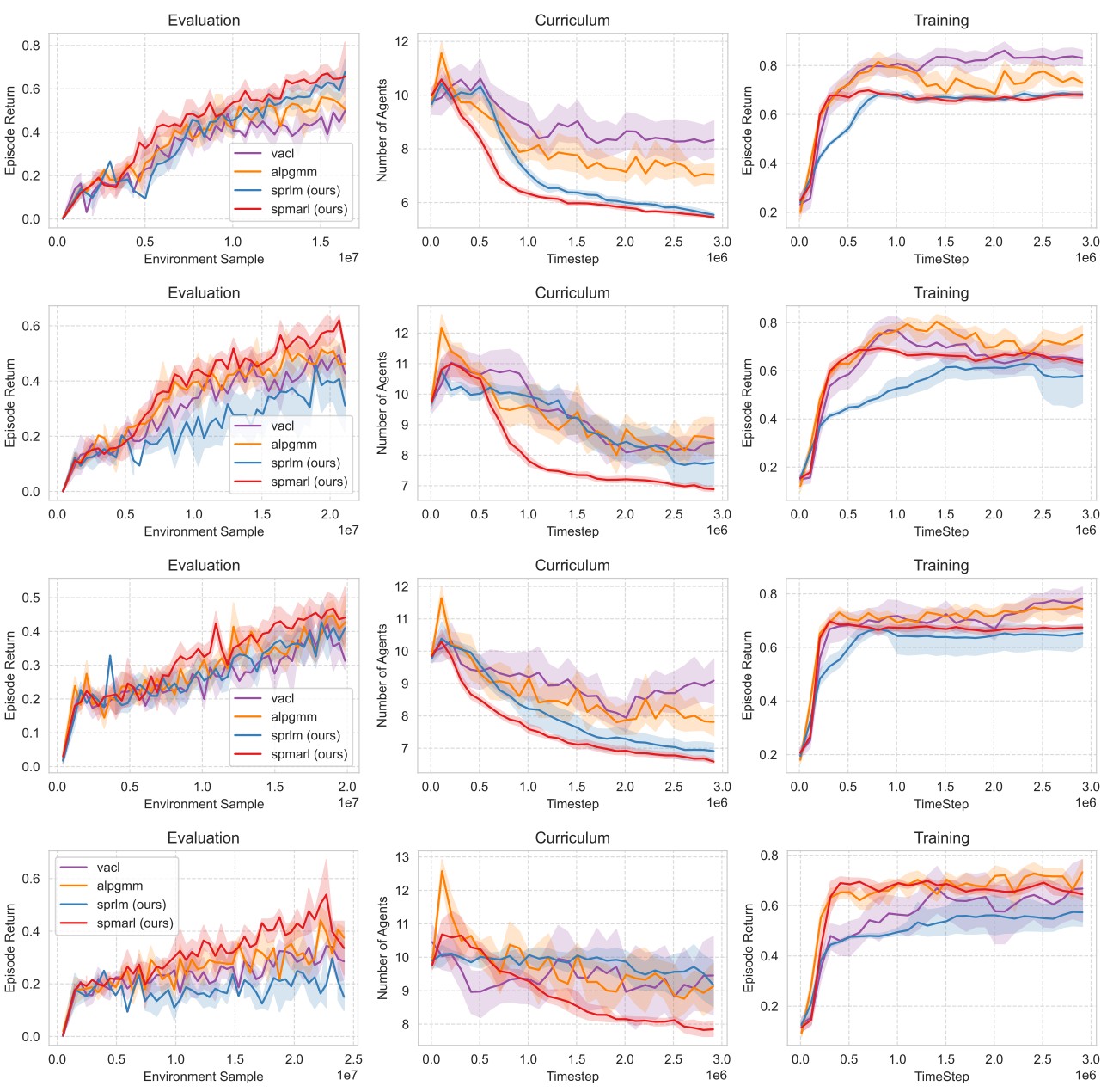

*Figure 8.* Comparison on SMACv2 *Terran* and *Zerg* tasks. From top to bottom row, the tasks are *Terran 5 vs. 5 Terran 6 vs. 6*, *Zerg 5 vs. 5*, and *Zerg 6 vs. 6*. Across all four tasks, SPMARL achieves performance that is comparable to or better than all baseline methods.

we empirically set $V_{\text{LB}}$ to approximately $80\%$ of the final task performance. For instance, in the SMACv2 *Protoss 5 vs. 5* task, where the final performance is around 0.8, we choose $V_{\text{LB}} = 0.6$. To evaluate the sensitivity of our method to this hyperparameter, we conduct an ablation study. The results, shown in Figure 11, demonstrate that our method performs robustly across a wide range of $V_{\text{LB}}$ values.

### A.4. Hyper-Parameters

We list the hyper-parameters of both the underlying multi-agent reinforcement learning algorithm MAPPO and our curriculum learning algorithm SPMARL. Note that we always set the same hyper-parameters for both SPMARL and SPRLM. The

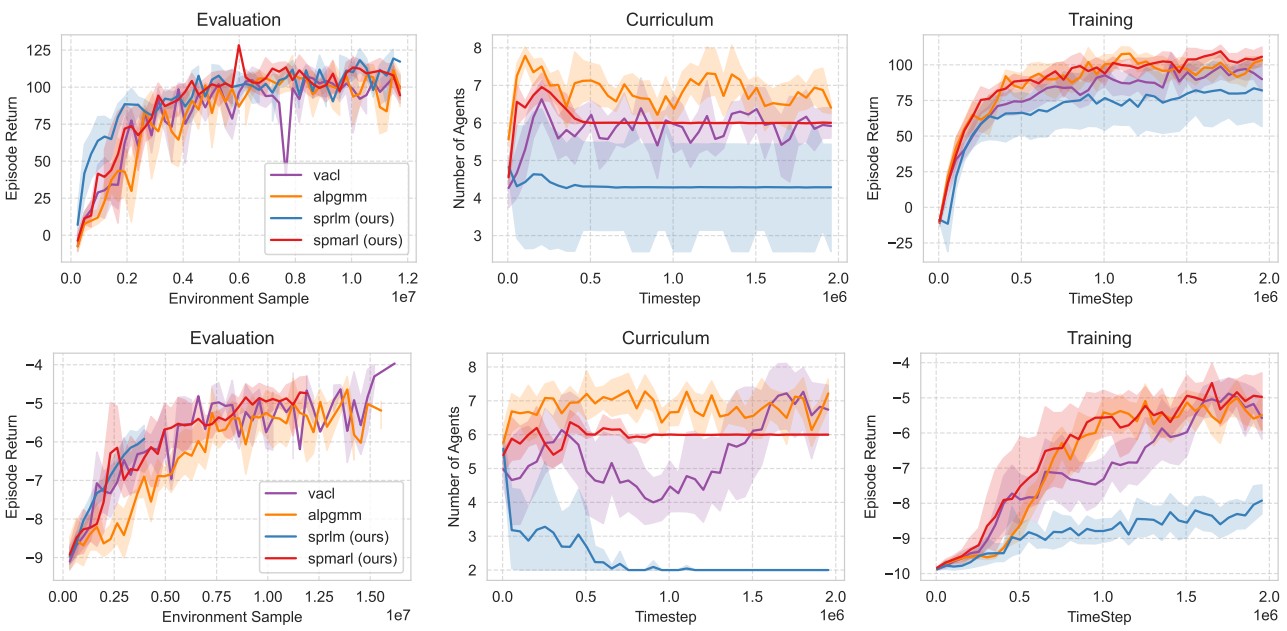

*Figure 9.* Comparison on *BenchMARL Balance* and *Wheel* tasks. From top to bottom, the tasks are *Balance* and *Wheel*. On these two **non-sparse** tasks, SPMARL shows comparable performance as other baselines, which could be due to that these tasks do not present exploration challenges. However, note that SPMARL is the only method generating curriculum that converges to target distributions.

shared parameters across all the domains are listed in Table 3 for MAPPO and Table 4 for SPMARL. Benchmark-specific parameters are listed in Table 5 for MPE, Table 6 for XOR game, Table 7 for SMAC v2, and Table 8 for *BenchMARL*.

*Table 3.* Common hyper-parameters of MAPPO across all domains

| Parameter | Value |
| --- | --- |
| use recurrent neural network | True |
| recurrent data chunk length | 10 |
| gradient clip norm | 10.0 |
| hidden layer dim | 64 |
| gae lambda | 0.95 |
| gamma | 0.99 |
| optimizer | Adam |
| optimizer epsilon | 1e-5 |
| entropy coefficient | 0.01 |
| ppo-clip | 0.2 |

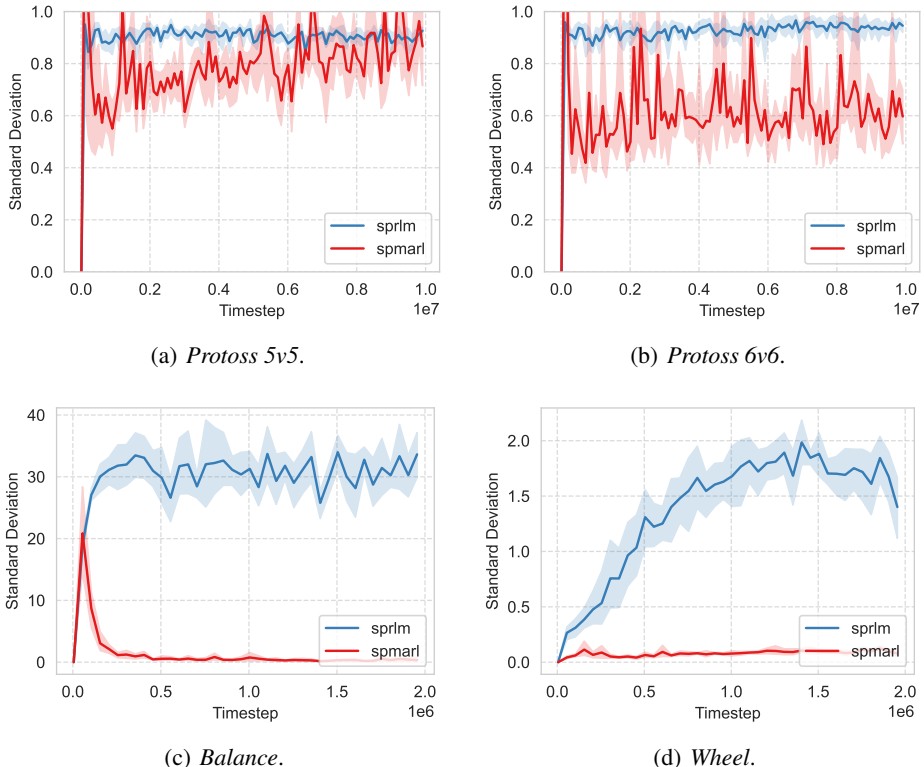

(a) *Protoss 5v5.*

(b) *Protoss 6v6.*

(c) *Balance.*

(d) *Wheel.*

*Figure 10.* Comparison of the standard deviation of the TD error objective used in SPMARL and that of the episode return used in SPRLM. The results show that the estimation variance of TD error used in SPMARL is usually lower than the variance of returns used in SPRLM.

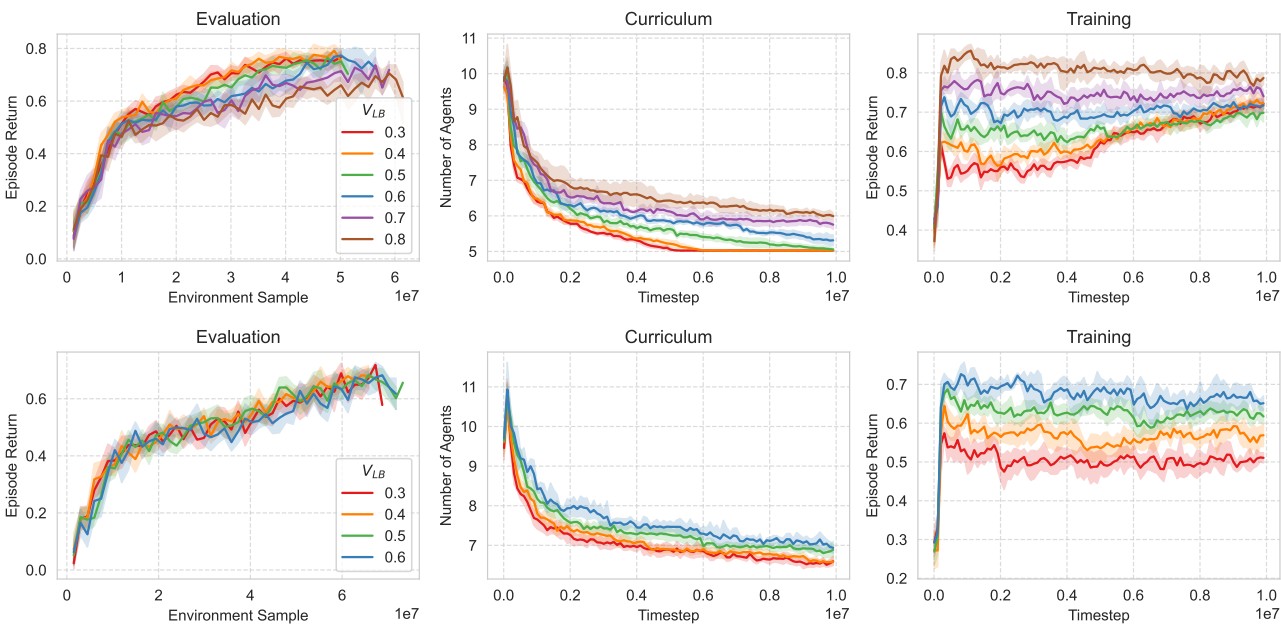

*Figure 11.* Ablation of $V_{\text{LB}}$ on SMACv2 *Protoss* tasks. From top to bottom, the tasks are *Protoss 5 vs. 5* and *Protoss 6 vs. 6*. The results indicate that SPMARL performs robustly across a broad range of $V_{\text{LB}}$.

*Table 4.* Common hyper-parameters of SPMARL across all domains

| Parameter | Value |
| --- | --- |
| max kl | 0.05 |
| context kl threshold | 8000 |
| target var | 4e-3 |
| std lower bound | 0.2 |

*Table 5.* Hyper-parameters for MPE

| Parameter | Value |
| --- | --- |
| MAPPO | |
| actor lr | 7e-4 |
| critic lr | 7e-4 |
| ppo epoch | 10 |
| hidden layer | 1 |
| episode length | 25 |
| number of mini batch | 1 |
| number of rollout threads | 256 |
| SPMARL | |
| lower context bound | 6 |
| upper context bound | 20 |
| init mean | 10 |
| init var | 25 |
| target mean | 8 |
| perf lb | 50 |

*Table 6.* Hyper-parameters for XOR

| Parameter | Value |
| --- | --- |
| MAPPO | |
| actor lr | 5e-4 |
| critic lr | 5e-4 |
| ppo epoch | 10 |
| hidden layer | 1 |
| episode length | 200 |
| number of mini batch | 1 |
| number of rollout threads | 50 |
| SPMARL | |
| lower context bound | 2 |
| upper context bound | 20 |
| init mean | 6 |
| init var | 16 |
| target mean | 20 |
| perf lb | 0.5 |

*Table 7.* Hyper-parameters for SMAC v2 tasks

| Parameter | Value |
|---|---|
| MAPPO | |
| actor lr | 5e-4 |
| critic lr | 5e-4 |
| ppo epoch | 5 |
| hidden layer | 1 |
| episode length | 400 |
| number of mini batch | 1 |
| number of rollout threads | 25 |
| SPMARL | |
| lower context bound | 5 |
| upper context bound | 15 |
| init mean | 10 |
| init var | 25 |
| target mean (*5v5*) | 5 |
| target mean (*6v6*) | 6 |
| target mean (*7v7*) | 7 |
| target mean (*8v8*) | 8 |
| perf lb (*Protoss 5v5, 6v6, 7v7*) | 0.6 |
| perf lb (*Protoss 8v8*) | 0.4 |
| perf lb (*Terran 5v5, 6v6*) | 0.5 |
| perf lb (*Zerg 5v5, 6v6*) | 0.5 |

*Table 8.* Hyper-parameters for *BenchMARL* tasks

| Parameter | Value |
|---|---|
| MAPPO | |
| actor lr | 5e-4 |
| critic lr | 5e-4 |
| ppo epoch | 5 |
| hidden layer | 1 |
| episode length | 200 |
| number of mini batch | 1 |
| number of rollout threads | 25 |
| SPMARL | |
| lower context bound | 2 |
| upper context bound | 10 |
| init mean | 5 |
| init var | 25 |
| target mean (*Balance*) | 6 |
| target mean (*Wheel*) | 6 |
| perf lb (*Balance*) | 80 |
| perf lb (*Wheel*) | -6 |

