# OpenReview forum: "Learning Progress Driven Multi-Agent Curriculum"
_ICML.cc/2025/Conference — ICML 2025 poster_

### Official Review · Reviewer_ffyd · 2025-03-12

**Overall Recommendation:** 2

**Summary:**

The authors apply an automatic curriculum design method, SPRL, to multi-agent reinforcement learning, using the number of agents as a parameter to control task difficulty. They call this method SPRLM.
They further extend SPRLM to maximise temporal difference error, which they call "learning progress", and call this algorithm SPMARL.
They evaluate SPRLM and SPMARL on 3 benchmarks against 2 different baselines.

**Claims And Evidence:**

The authors claim that their method "outperforms state-of-the-art baselines".
However, the results on their benchmarks are mixed - on the XOR task, they significantly improve on the baselines, on the SimpleSpread task, they slightly improve upon baselines, and on the the SMAC-v2 baseline performance is inline with one of their baselines. They provide mean and std of these results in the appendix.

**Essential References Not Discussed:**

The authors miss a discussion of the field of Unsupervised Environment Design (UED) which is a prominent direction of research in curriculum design and reinforcement learning.

**Experimental Designs Or Analyses:**

See above, I have concerns about the comprehensiveness of the benchmarks.

**Methods And Evaluation Criteria:**

The authors present results on 3 benchmarks, SimpleSpread, XOR and the Protoss task from SMAC-v2.
Whilst SimpleSpread is illustrative of a task in which more-agents does not make the task harder, it seems overly simple, as does XOR. SMAC-v2 has other tasks, Terran and Zerg, which are ignored.
I am not very familiar with MARL, but it is my understanding that more comprehensive benchmarks exist, such as JaxMARL or MARL Bench.

**Other Comments Or Suggestions:**

No further comments

**Other Strengths And Weaknesses:**

Strengths: Extending curriculum design in MARL to automatically set the number of agents is useful, and applying it a setting such a SimpleSpread that gets easier with more agents is clever. Additionally combining this method with TD error is a neat way to get around sparse rewards.

Weaknesses: The discussion of prior and related work is confusing - it is not clear to what extent the method presented in Section 4 is background or novel to their approach. Further, the actual explanation of the method is extremely difficult to follow, despite Section 4, Algorithm 1 and the diagram in appendix A.  The work would benefit from more time clarifying the background to this method and clarifying section 4 and algorithm 1.

As above, I am concerned with the suitability of the benchmarks.

**Questions For Authors:**

Are there better benchmarks you can add to this?

**Relation To Broader Scientific Literature:**

The authors cite 2 papers from 2022, 1 from 2023 and 1 from 2024. All other citations are pre-2022. Whilst I am not expert in MARL, I struggle to believe such little relevant work has been done on curriculum learning in MARL over the last few years.

**Theoretical Claims:**

No theoretical claims.

---

> ### Author Rebuttal · Authors · 2025-04-01
>
> ## Response to Reviewer ffyd
> We thank the reviewer for the valuable comments. We note that the main concern comes from the comprehensiveness of the benchmarks and inclarity of the method description. In response, we have conducted additional experiments on new benchmarks, and we hope that the enhanced clarity in our revisions will effectively address these concerns.
>
> ### 1. More SMACv2 tasks and BenchMARL:
>
> We have conducted additional experiments on four new SMACv2 tasks: *Terran 5v5*, *Terran 6v6*, *Zerg 5v5*, and *Zerg 6v6*. The results, available at this [link](https://sites.google.com/view/spmarl-icml2025/#h.bj45hu212vx7), demonstrate that our algorithm, SPMARL, consistently outperforms the baselines and generates stable, interpretable curricula across all tasks.
>
> For the BenchMARL tasks, *Balance* and *Wheel*, our method achieves performance comparable to the baselines, as shown in this [link](https://sites.google.com/view/spmarl-icml2025/home#h.oawhzhwh26av). This may be due to the fact that the exploration challenge in these tasks is not particularly severe, reducing the necessity of curriculum learning. However, we note that SPMARL is the only one generating curriculum that converges to the target task distribution.
>
>
> ### 2. Clarity on Section 4, Algorithm 1:
> We apologize for the lack of clarity in explaining our method. In Section 4, we primarily apply the existing single-agent curriculum learning method, SPRL, to control the number of agents, which we refer to as SPRLM. Our main contribution is the introduction of the TD-error-based learning progress, a novel approach to curriculum learning, as noted by reviewer ZwRM.
>
> Intuitively, our method first identifies tasks, defined by varying numbers of agents, that exhibit the highest learning progress. After training on these high-progress tasks and achieving the performance threshold $V_{LB}$, we adjust the task distribution towards the target by minimizing the $KL$ divergence between current task distribution and the target distribution. This process is constrained by the $KL$ divergence between the old and new task distributions to prevent rapid changes in the task distribution.
>
> We appreciate the reviewer's valuable suggestions on the organization of the paper. We agree that moving the diagram and comparison table from the appendix to the main text will improve readability. We will revise the paper accordingly based on your helpful recommendations.
>
> ### 3. Related literature
>
> We appreciate the reviewer's suggestion regarding the literature on Unsupervised Environment Design (UED), which falls within the domain of single-agent curriculum learning. UED approaches, such as [1], primarily focus on general environment design using state-action coverage as the objective. In contrast, our work addresses the increased credit assignment difficulty in multi-agent reinforcement learning (MARL) when applying reward-based methods. We acknowledge the close relationship between these fields and will update our paper to include a discussion on this topic.
>
> Compared to single-agent RL, research on curriculum learning in MARL has seen significantly less progress. This is partly because many single-agent curriculum learning methods, such as SPRL and VACL, can be adapted to MARL to control the number of agents. However, our work specifically investigates the challenges that arise in MARL and proposes a TD-error-driven curriculum tailored for MARL.
>
> [1] Teoh, Jayden et al. "Improving Environment Novelty Quantification for Effective Unsupervised Environment Design." Advances in Neural Information Processing Systems 37 (2024).

---

### Official Review · Reviewer_p7uW · 2025-03-14

**Overall Recommendation:** 2

**Summary:**

The paper presents a curriculum learning method for MBRL, where the task difficulty is controlled by the number of agents, using TD error for learning progress measurement. The method is evaluated on three sparse-reward benchmarks and presents empirical advantages over baselines.

### Update after rebuttals
Thank you authors for the rebuttals. I have two further concerns after reading the rebuttals and other reviewers' comments:
1. In the rebuttal, the author mentioned that $V_\text{LB}$ is chosen heuristically according to the converging performance, which is unknown before running the algorithm. It raises questions on how to apply these heuristics in a new environment.
2. The additional experiments in response to reviewer ffyd show that the advantage of the proposed method compared to prior arts is much smaller compared to environments shown in the original manuscript.
3. The authors also did not provide the action and state space specifications for these environments, making it hard for readers to interpret the advantage of the method in complex environments.

For the reasons above, while agent number adjustment is worth investigating for MBRL, I'd maintain my score for the paper in its current form.

**Claims And Evidence:**

The paper claims that using TD error for learning progress reduces variance of objective estimations, and supports the claim using Figure 8 and 10. However, the settings of these two figures can be further explained, including details such as how many samples the variance is computed over. Given that this is a core claim of the paper, analysis on more environments should be shown.

**Essential References Not Discussed:**

Prior works on intrinsic reward driven curriculum learning, e.g. curiosity-driven learning [1], are not discussed.

[1] Pathak, D., Agrawal, P., Efros, A. A., & Darrell, T. (2017). Curiosity-driven Exploration by Self-supervised Prediction.

**Experimental Designs Or Analyses:**

The method is evaluated across 3 benchmarks, each across several seeds.

**Methods And Evaluation Criteria:**

The method is evaluated on selected benchmarks.

**Other Comments Or Suggestions:**

* Environment configurations, including state space and action space, should be specified.

**Other Strengths And Weaknesses:**

Weaknesses:
* The key hyperparameter, $V_\text{LB}$, is not ablated. How to choose hyperperparameter for new environments?

**Questions For Authors:**

1. How many seeds are the experiments in Fig. 3-4 evaluated on?

**Relation To Broader Scientific Literature:**

The paper presents novelty by leveraing the number of agents to control task difficulties and using TD errors for learning progress estimation.

**Theoretical Claims:**

No theoretical proofs.

---

> ### Author Rebuttal · Authors · 2025-04-01
>
> ## Response to Reviewer p7uW
> We thank the reviewer for the appreciation of the novlty of our method. We hope our clarification and new experiments help to address your concerns.
>
> ### 1. Clarity on the variance comparison:
> > However, the settings of these two figures can be further explained, including details such as how many samples the variance is computed over. Given that this is a core claim of the paper, analysis on more environments should be shown.
>
> In Figures 6 and 8, we simply compute the standard deviation of the collected episode returns and TD errors in each iteration, which are collected with the same context samples and used to estimate the objectives of the curriculum learning methods. The number of samples equals to the number of episodes done in the iteration which is at least $25$ episodes in *SMACv2* since some episodes may terminate earlier.
>
> Mathematically, assume we have collected a set of contexts $\\{c_1, c_2, \dots, c_n\\}, n \approx 25$, the corresponding episode returns $\\{R_{1}, R_{2}, \dots, R_{n}\\}$, and the TD-errors $\\{TD_{1}, TD_{2}, \cdots, TD_{n}\\}$ where $TD_{i} = LP(c_i) = \\frac{1}{2} E_{s, a \sim \pi(a \vert s, c_i)} \left[ \lVert R(s, \mathbf{a}) - V(s) \rVert^2 \right]$, the standard deviation is computed as $\sigma = \sqrt{\frac{1}{n} \sum_{i=1}^{n} (x_i - \bar{x})^2}$, where $x_i$ can be $R_{i}$ or $\text{TD}_i$ and $\bar{x}$ represents the mean. Since TD-error on each context has been averaged over the samples of the whole episode, it shows lower estimation variance.
>
> We ran additional experiments and performed similar analysis on two tasks from BenchMARL, i.e. *Balance* and *Wheel*. The results in [link](https://sites.google.com/view/spmarl-icml2025/home#h.cnlpmxrzpuhe) show that our method SPMARL demonstrates much lower standard deviation than the return estimation used in SPRL. In these experiments, the samples are also at least $25$ for each computation of the standard deviation, sometimes it can be more than $25$ when some episodes terminate earlier.
>
> The variance reduction of using TD error compared to episode returns is also thoroughly analyzed in [1].
>
> [1] Schulman, John, et al. "High-Dimensional Continuous Control Using Generalized Advantage Estimation." Proceedings of the International Conference on Learning Representations (ICLR), 2016.
>
>
>
> ### 2. Clarity on choosing $V_{LB}$:
> >The key hyperparameter, $V_{LB}$, is not ablated. How to choose hyperperparameter for new environments?
>
> Thanks for raising this issue. We are sorry for not explaining it clearly. $V_{LB}$ is an important hyperparameter in our method. We empirically choose $V_{LB}$ to be $80\%$ of the final performance. For example, in Protoss 5v5, the converged win rate is around 0.8, we set $V_{LB}$ with 0.6. However, our ablation in [link](https://sites.google.com/view/spmarl-icml2025/home#h.g3dhqeny6kze) shows that $V_{LB}$ can be chosen over a larger range.
>
>
> ### 3. Number of random seeds:
> > How many seeds are the experiments in Fig. 3-4 evaluated on?
>
> We appreciate the reviewer's feedback in highlighting this ambiguity. All our experiments were conducted using five random seeds.
>
> ### 4. Discussion on curiosity-driven learning:
>
> Thank you for your insightful comment. Curiosity-driven exploration is a well-studied topic in RL, primarily focusing on developing intrinsic reward signals through techniques such as self-prediction, rather than modifying environments as in curriculum learning. However, we believe that general curiosity-driven exploration can be effectively integrated with curriculum design to further enhance exploration.
>
> ### 5. Clarity on state, action space
>
> We thank the reviewer for raising this issue. In the appendix, in the next paper version, we include a detailed introduction of all the tasks used in our experiments.

---

### Official Review · Reviewer_ZwRM · 2025-03-26

**Overall Recommendation:** 3

**Summary:**

This paper looks at curriculum learning in multi-agent reinforcement learning (MARL) by using the number of agents as a dynamic context variable. The authors first adapt self-paced reinforcement learning (SPRL) to the multi-agent setting (SPRLM), then propose SPMARL - a more principled variant that measures learning progress via TD error rather than noisy episode returns. Across several sparse-reward MARL benchmarks, SPMARL shows faster convergence and stronger final performance than prior approaches. Benchmarks are sufficient but not extensive, leaving out some of the JaxMARL work and others that have appeared in the last couple of years. The simplicity of the method is very valuable, though, given the complexity of MARL environments and the MARL problem writ large.

**Claims And Evidence:**

If I'm understanding the paper correctly, the setting is fully observable (albeit decentralized).  Full observability feels like a potential hitch.  In a rebuttal, would love to hear the authors' thoughts on that.

**Essential References Not Discussed:**

See experimental section - there are outdated / missing references from the last eighteen or so months.

**Experimental Designs Or Analyses:**

MARL research, for better or for worse, hinges on experimental results at the moment.  It's not a negative that there aren't theoretical results in this paper, that's the norm.  That said, the experimental analysis seems to leave off some recent tools for benchmarking like JaxMARL and others.  I am empathetic to how expensive these experiments can be to run, but it's still good to keep up to date on what's out there in a fast-moving field.

**Methods And Evaluation Criteria:**

Sparse exploration is hard, and I appreciate the authors focusing on that. The framework also feels like it could be super general and applied across single-agent RL and MARL.

**Other Comments Or Suggestions:**

N/A

**Other Strengths And Weaknesses:**

N/A

**Questions For Authors:**

See above.

**Relation To Broader Scientific Literature:**

See experimental section - there are outdated / missing references from the last eighteen or so months.

**Theoretical Claims:**

N/A

---

> ### Author Rebuttal · Authors · 2025-04-01
>
> ## Response to Reviewer ZwRM
> We thank the reviewer for the appreciation of the simplicity and generality of our method. We hope new experiments and clarifications help to address your concerns.
>
> ### 1. Clarity on observability setting:
> >If I'm understanding the paper correctly, the setting is fully observable (albeit decentralized). Full observability feels like a potential hitch. In a rebuttal, would love to hear the authors' thoughts on that.
>
> We agree with the reviewer that observability is important in MARL. In our experiments, we use partial observability for all the tasks except the *XOR* matrix game, since the *XOR* game is a stateless task. For example, in *Simple-Spread*, the agents can only observe $4$ nearest agents and landmarks, and in *SMACv2* tasks agents can only observe enemies and agents in a certain range. Partial observability also helps in transferring policies across tasks with different numbers of agents since the observation space is fixed instead of growing with the number of agents. We use RNN policies for all the methods.
>
>
> ### 2. More benchmarks:
> > The simplicity of the method is very valuable. ... The framework also feels like it could be super general and applied across single-agent RL and MARL.
> > That said, the experimental analysis seems to leave off some recent tools for benchmarking like JaxMARL and others.
>
> We appreciate the reviewer for kindly acknowledging the contribution of our simple yet effective method and thank you for suggesting new benchmarks. We found that our current benchmarks such as *MPE simple-spread* and *SMACv2* are also included in *JaxMARL*. Therefore, we mainly performed new experiments on more *SMACv2* tasks and two tasks from *BenchMARL* [1] suggested by reviewer ffyd.
>
> The results in [link](https://sites.google.com/view/spmarl-icml2025/#h.bj45hu212vx7) show the results on four new *SMACv2* tasks, *Terran 5v5*, *Terran 6v6*, *Zerg 5v5*, and *Zerg 6v6*. We can see that our algorithm SPMARL consistently outperfoms the baselines and generates stable and intepretable curricula across all the tasks.
>
> For the BenchMARL tasks, *Balance* and *Wheel*, our method achieves performance comparable to the baselines, as shown in this [link](https://sites.google.com/view/spmarl-icml2025/home#h.oawhzhwh26av). This may be due to the fact that the exploration challenge in these tasks is not particularly severe, reducing the necessity of curriculum learning. However, SPMARL is the only one generating curriculum that converges to the target task distribution.
>
> [1] Bettini, Matteo, Amanda Prorok, and Vincent Moens. "Benchmarl: Benchmarking multi-agent reinforcement learning." Journal of Machine Learning Research 25.217 (2024): 1-10.

---

### Decision · Program_Chairs · 2025-05-01

**Decision:**

Accept (poster)

**Comment:**

This paper looks at curriculum learning in MARL.  Curriculum learning in the *single*-agent setting, roughly, breaks a task into subtasks and then generates a (fungible, changeable) curriculum over those subtasks to accomplish the task.  In the *multi*-agent setting, which is what this paper looks at, the problem is at a high level the same but the curriculum chosen by one agent might influence those chosen in later stages by other agents, and so on - that is, some form of adversarial reasoning enters into the picture.  This is a complex environment.  This paper posits to add the *number* of agents into the curriculum learning problem as a variable.  That feels natural to this AC (it’s sort of a proxy for the complexity of one important part of the environment), so is worth exploring.  Moving in deeper, the paper looks into credit assignment as the number of agents (that primary variable for CL) changes, and at reducing the variance in the estimate of the value of return ablated across number of agents.

Let this AC start by stating that experimental design in RL is hard, MARL is harder, and doing CL as a function of the number of agents in MARL is even harder.  Lots of empathy there.  That said, let’s bring up some very valid points from reviewers:
- Related work, and discussion therein (all reviewers, especially ffyd and [7uW) - there is not a lot of placement of this work in the current literature, and we’d all appreciate a better discussion there.  This AC is not an expert in CL for MARL, but is well-versed in CL for single-agent RL as well as broadly versed in MARL.  Is this *really* the full set of works that have been published in this space in recent years?  I find it hard to believe that like five papers total have been published in this space in recent years, but after some searching I am inclined to believe that is correct.  If this is the response a set of semi-experts in the area are having to the paper, it’d be good to fix that in the text.
- How do JaxMARL and BenchMARL fit into all of this, *explicitly*?  This AC sees the authors’ rebuttal to ZwRM bringing this up.  Clarify what’s going on, and if there are gaps in a future version of the paper definitely compare against those.  These are becoming standard MARL benchmark suites.
- Questions of generalization around the V_LB parameter - this AC encourages you to add further ablation studies here specifically looking across environments (i.e., generalization) in a future version of the paper.